# Development and validation of a multiplex immunoassay for the simultaneous quantification of type-specific IgG antibodies to E6/E7 oncoproteins of HPV16 and HPV18

Hans Layman[1]*, Keith W. Rickert[2], Susan Wilson[2], Anastasia A. Aksyuk[3], Jill M. Dunty[3], Dusit Natrakul[3], Nithya Swaminathan[1,¤a], Christopher J. DelNagro[1,¤b]

1 AstraZeneca plc, South San Francisco, California, United States of America, 2 AstraZeneca plc, Gaithersburg, Maryland, United States of America, 3 Meso Scale Diagnostics, LLC., Rockville, Maryland, United States of America

☮ These authors contributed equally to this work.
¤a Current address: Gilead Sciences, Inc. Foster City, California, United States of America
¤b Current address: Exelixis, Inc., Alameda, California, United States of America
‡ These authors also contributed equally to this work.
* hans.layman@astrazeneca.com

## Abstract

More than 170 types of human papilloma viruses (HPV) exist with many causing proliferative diseases linked to malignancy in indications such as cervical cancer and head and neck squamous cell carcinoma. Characterization of antibody levels toward HPV serology is challenging due to complex biology of oncoproteins, pre-existing titers to multiple HPV types, cross-reactivity, and low affinity, polyclonal responses. Using multiplex technology from MSD, we have developed an assay that simultaneously characterizes antibodies against E6 and E7 oncoproteins of HPV16 and 18, the primary drivers of HPV-associated oncogenesis. We fusion tagged our E6 and E7 proteins with MBP via two-step purification, spot-printed an optimized concentration of protein into wells of MSD 96-well plates, and assayed various cynomolgus monkey, human and HPV+ cervical cancer patient serum to validate the assay. The dynamic range of the assay covered 4-orders of magnitude and antibodies were detected in serum at a dilution up to 100,000-fold. The assay was very precise (n = 5 assay runs) with median CV of human serum samples ~ 5.3% and inter-run variability of 11.4%. The multiplex serology method has strong cross-reactivity between E6 oncoproteins from human serum samples as HPV18 E6 antigens neutralized 5 of 6 serum samples as strongly as HPV16 E6. Moderate concordance (Spearman's Rank = 0.775) was found between antibody responses against HPV16 E7 in the multiplex assay compared to standard ELISA serology methods. These results demonstrate the development of a high-throughput, multiplex assay that requires lower sample quantity input with greater dynamic range to detect type-specific anti-HPV concentrations to E6 and E7 oncoproteins of HPV16 and 18.

**Data Availability Statement:** All relevant data are within the manuscript and its Supporting Information files.

**Funding:** HL, KAR, SW, NS, CJD are current or former employees of AstraZeneca. They received compensation in the form of salary and stock as employees. AAA, JMD, and DN are current or former employees of Meso Scale Diagnostics LLC and receive compensation in the form of salary and stock. AstraZeneca nor Meso Scale Diagnostics LLC did not play a role in the study design, data collection and analysis, decision to publish, or preparation of the manuscript and only provided financial support in the form of salaries and research materials.

**Competing interests:** HL, KAR, SW, NS, CJD are current or former employees of AstraZeneca. They received compensation in the form of salary and stock as employees. AAA, JMD, and DN are current or former employees of Meso Scale Diagnostics LLC and receive compensation in the form of salary and stock. The commercial affiliations of AstraZeneca and Meso Scale Diagnostics LLC does not alter any adherence to PLOS ONE policies on sharing data and materials.

# 1. Introduction

Human papilloma virus (HPV) is a diverse DNA virus from the papillomavirus family with 170 types known implicated in numerous etiologies including sexually transmitted diseases, skin lesions, and cancers [1–3]. At least 12 of these HPV types are high-risk as persistent HPV infection has been linked to cancers of the oropharynx, larynx, vulva, vagina, cervix, and anus [2,3]. HPV-associated cancers make up over 5% of all cancer cases diagnosed yearly and has been increasing since 2006 [4,5]. Of interest, oral infection with HPV16 and 18 account for a significant proportion of head and neck cancer (HNSCC) patients worldwide [6]. More specifically, oncoproteins E6 and E7 of HPV16 and 18 are of great importance as they drive suppression of pRb and increase levels of p16 protein via negative feedback mechanisms driving tumor progression [7,8]. Seropositivity of type-specific IgG polyclonal antibodies to these HPV oncoproteins in patients has demonstrated improved overall survival in HNSCC [9], therefore we sought to develop an assay that can evaluate antibodies to HPV16/18 E6 and E7 oncoproteins.

Assays employing analysis of antibodies to E6/E7 oncoproteins to HPV16 and 18 have demonstrated mixed results with few groups showing concordance between HPV16/18 genotype versus serotype positive typically relying on p16 genotype positivity [9–13]. These results may be attributed to the assembly patterns with E6 proteins [8] as E6 commonly folds and aggregates hindering the ability and accessibility of antibodies to bind *in vitro* [14]. To overcome this attribute, many groups have leveraged pseudovirion incorporation [15,16] or protein fusion tags such MBP or GST to reduce protein aggregation [17] resulting in robust and reproducible HPV serology assays toward serotyping of individuals in clinical trials. Here, we have developed MBP tagged versions of E6 and E7 oncoproteints of HPV16 and 18 for detection of anti-HPV16/18 positive sera.

Conventional assays for serology such as ELISA allow only one antigen at a time to be evaluated per well. Various multiplexing technologies for antibody-protein interactions have been validated for clinical use [18]. Competitive Luminex Immunoassays (cLIA) is a bead-based technology that has been leveraged in numerous HNSCC biomarker studies with mixed results attributed to modest increases in dynamic range compared to ELISA, lack of standardized reagents and lack of uniform methods to establish cut-off values [16]. Furthermore, chemical linkers to conjugate beads to protein may result in protein conformational changes [15]. Meso Scale Diagnostics, LLC (MSD, Rockville, MD) uses electrochemiluminescent (ECL) technology lending itself to a broad dynamic range. Spots for an MSD$^{\circledR}$ ECL assay are printed in array and all antibodies of interest can be analyzed in one well. The small area of spots requires significantly less coating protein and sample reducing the quantity of consumables necessary to perform serologic analysis in complex clinical trials. Additionally, the proteins coated on the plate surface maintain their conformational state. Herein, we have developed and report a method for multiplex HPV serology analysis of antibodies specific to MBP-tagged HPV16 and 18 E6/ using the MSD platform. We investigated key assay performance metrics toward type-specific anti-HPV antibody concentrations including precision, linearity, reproducibility, robustness, cross-reactivity, and correlation to conventional ELISA across serum from suspected HPV+ adult donors, pediatric donors, and HPV16/18 E6 and E7 vaccinated cynomolgus monkeys.

# 2. Materials and methods

## 2.1 HPV16/18 E6 and E7 antigen production and purification

Protein sequences for HPV16/18 E6 and E7 were derived from UniProt (S1 Fig) and cloned into the pMalC5X vector facilitating the fusion and allowing the production of MBP-HPV16/

18 E6 and MBP-HPV16/18 E7 proteins. All constructs were expressed in *Escherichia coli* BL21D3 cells (New England BioLabs), grown in LB medium supplemented with kanamycin (50 μg/mL) at 30˚C until $OD_{600}$ = 3.0 followed by a temperature reduction to 20˚C and addition of IPTG at 1mM to the culture. Cells were harvested by centrifugation and lysed with BPER. Lysates were subjected to SDS-PAGE and anti-His Western blot.

MBP-tagged HPV16 E6/E7 and HPV18 E6/E7 were bound to a 5 mL MBPtrap column (GE Healthcare) in 20 mM Tris, 400 mM NaCl at pH 7.5 and eluted with 10 mM maltose. This was followed by dialysis and further purification on a Q-HP column at pH 8, eluting with a gradient from 0–500 mM NaCl. Size exclusion chromatography (SEC) and tandem mass spectrometry (MS) was run on proteins to confirm column elution time and identification of predicted proteins.

## 2.2 Antibodies, serum samples, and reference serums

Human samples were obtained from BioIVT (Hicksville NY) or Proteogenex (Inglewood, CA) with informed consent. Monkey serum samples were obtained as a gift from Jean Boyer and the animal protocol was approved with IACUC Board approval at Inovio Pharmaceuticals in San Diego, CA.

For assay development and validation, a heat-inactivated pool of human serum positive for anti-HPV16/18 E6 and E7 was used as a reference. Serum from various human donor cohorts was obtained with Ethics committee approval and informed consent obtained from BioIVT (Hicksville, NY) normal healthy adults (n = 6), adults with ≤1 sexual partner (n = 30), pediatric donors (age 1–5 days; n = 10), HPV+ serum from cervical cancer patients at CIN2/3 staging (Proteogenex Inglewood, CA; n = 3), and serum of HPV16/18 E6 and E7 vaccinated cynomolgus monkeys (n = 6; gifted by Dr. Jean Boyer, Inovio Pharmaceuticals, San Diego, CA) were used as samples to evaluate assay suitability. IgG depleted serum (BBI Solutions, Crumlin, UK) was used as a negative control in the assay. Type-specific mouse monoclonal antibodies to HPV16/18 E6 (Clone C1P5, Santa Cruz Biotechnology, Santa Cruz, USA), HPV16 E7 (Clone 716–281, ThermoFisher Scientific, South San Francisco, USA), HPV18 E6 (Clone G-7, Santa Cruz Biotechnology, Santa Cruz, USA), and HPV18 E7 (Clone F-7, Santa Cruz Biotechnology, Santa Cruz, USA) were purchased for specificity experiments.

## 2.3 MSD multiplex type-specific anti-HPV16/18 E6 E7 ECL protocol

Serological immunoassays with electrochemiluminescence (ECL) detection were performed using instrumentation and multiwell plate consumables from MSD.Assay plates were coated with an optimized concentration of HPV Concentration of antigens coated in well are 300 μg/mL for MBP-HPV16 E6, 400 μg/mL for MBP-HPV16 E7 and MBP-HPV18 E6, and 200 μg/mL for MBP-HPV18 E7. Plates were equilibrated to room temperature (RT) for 30 minutes prior to use and serum samples thawed on ice. 150 μL of MSD blocker A is added to the plate and incubated at RT for 1 hour with shaking at 705 rpm. Plates are washed three times with PBS-T followed by adding 50 μL of reference standard or sample to the wells at a 500-fold dilution in duplicate in assay diluent (Diluent 100, MSD). Samples were incubated for 2 hours at RT with shaking at 705 rpm. Following sample incubation, plates were washed 3 times with PBS-T. 50 μL of mouse monoclonal anti-Human IgG detection antibody labelled with SULFO-TAG™ label at a working concentration of 2 μg/mL was added to all wells of the plate and incubated at RT for 1 hour with shaking. Following incubation in detection antibody, plates were washed three times in PBS-T, MSD GOLD™ Read Buffer was added to the plate at a volume of 150 μL/well and the plate was read. The measurement of samples was performed on a SECTOR S 600 instrument and raw ECL values were acquired. For each antibody, the ECL

signal was converted to AU/mL by interpolation from a four-parameter logistic standard curve based on the 4-fold, 8-point serial dilution (100–409,600-fold dilution) of the pooled human reference serum.

## 2.4 Establishing assay performance characteristics

**2.4.1 Assay specificity.** The specificity of the HPV multiplex ECL assay was assessed through evaluation of antigen-specific antibodies (n = 4) and their ability to be detected based on other coated spots in the well. Wells were treated as previously described with the addition of mouse monoclonal antibodies at the described concentrations: anti-HPV16/18 E6 (25 ng/mL), anti-HPV16 E7 (4 ng/mL), anti-HPV18 E6 (4 ng/mL), and anti-HPV18 E7 (1 ng/mL) followed by detection with MSD goat anti-mouse SULFO-TAG labelled detection antibody.

Specificity was confirmed using a competition assay by incubating the synthesized HPV soluble antigens with mouse monoclonal antibodies, as well as 4 adult donor serum samples and 2 HPV+ cervical cancer patient serum samples. Briefly, antibodies were pre-diluted to the concentrations described previously and samples were pre-diluted 500-fold. Samples were spiked with individual soluble antigens at 10x the coating mass. Samples were pre-incubated for 1 hr to facilitate neutralization before adding to the assay plate. The assay was conducted as described before and the ability to detect an antibody's specificity and cross reactivity was expressed as a percent of on-board diluent spiked ECL signal.

**2.4.2 Performance of reference serum.** Two adult donors were identified with high antibody concentrations to HPV16/18 E6 and E7, pooled, and the performance of the reference serum was evaluated from 4-fold serial dilutions (100–409,600 fold) of serum in the multiplex assay (n = 5 assay runs). From these runs, a concentration reference assignment was assigned to each analyte across the dilution range.

**2.4.3 Validation of the type-specific anti-HPV antibody assay.** To validate the assay, the dilutional linearity, precision, range, and limits of detection were assessed over a range of serum samples: adult donor, HPV+ cervical cancer, pediatric, and monkey. In addition, the stability of samples in the assay was assessed after multiple freeze/thaw cycles and different sample storage conditions.

The dilutional linearity of the assay was established using 8 different human serum samples, and the reference serum. The human sera were tested at 2-fold dilution steps over 8 concentrations (100–12,800-fold dilution) to capture the dynamic range of the assay. The results of the antibody concentrations were normalized to the observed concentration at an 800-fold dilution and incorporated in semi-log plots.

The upper (ULOQ) and lower limit of quantification (LLOQ), defined as the maximal and minimal concentration of polyclonal anti-human HPV antibodies that is reliably quantified with %CV <20% and recovery between 80–120%, was determined for the assay. To quantify the ULOQ, reference serum was diluted 100-fold into assay diluent followed by three 1.2-fold dilutions. To quantify the LLOQ, two LLOQ sample series were prepared by spiking reference serum into IgG depleted serum, series one was pre-diluted 10-fold, and series two was pre-diluted 50-fold with the standard on-board dilution of 500-fold for standard sample analysis.

The precision of the assay (n = 6 assay runs) was established using 7 adult donor serum samples, 10 pediatric serum samples (suspected low concentration), and IgG depleted serum. Evaluation of the intra- and inter-assay calculated sample concentrations to each HPV antigen and the variance associated with the measurement to establish assay precision.

**2.4.4 Comparison with an ELISA type-specific anti-HPV antibody assay.** MSD multiplex and a standard single-plex ELISA for HPV16 E7 were compared head-to-head to confirm anti-HPV16 E7-specific responses and to determine the assay's dynamic range. An additional

subset of serum samples: 16 normal healthy donors, and 10 HPV+ cervical cancer serum samples were obtained from BioIVT or Proteogenex and tested across 8 concentrations of serum at 4-fold dilutions (10–163,480-fold dilution, on board). To evaluate the correlation between MSD and ELISA anti-HPV16 E7 signals, Spearman's Rank Order Correlation Coefficient was utilized to confirm the relationship between the two technologies as the linear relationship between them is monotonic in reporting of ECL signal to $OD_{450}$ values across all dilutions of each sample. In analysis of the semi-log data (log ECL signal versus $OD_{450}$ ELISA), a Pearson coefficient is used to calculate linear-transformed data to confirm correlation between MSD and ELISA.

## 3. Results

### Generation of HPV antigens

In order to overcome the hinderance of aggregation-prone E6 proteins, we evaluated the suitability of linking E6 or E7 proteins to maltose binding protein to reduce aggregation for improved reactivity in downstream assays. In evaluating the SDS-PAGE gel of MBP-HPV18 E6, we observed our expected mass of protein (MBP-HPV18 E6) at 64 kDa, however, substantial impurities in purification were observed (Fig 1A) before (lanes 4 and 5) and after reduction (lane 8) with DTT. SDS-PAGE gel evaluation following QHP purification (Fig 1B) demonstrated that the major impurities flowed through the column (lanes 2 and 3). Size-exclusion chromatography (SEC) demonstrated that a vast majority of the DTT reduced, peak MBP-HPV18 E6 protein is an aggregate with an estimated mass of $4x10^6$ Da that elutes between 5 and 6 minutes (Fig 1C). The purified MBP-HPV18 E6 protein was analyzed by mass spectrometry (Fig 1D) and the primary mass observed is that of the predicted, intact protein at 63589 kDa, demonstrating that the observed aggregation is non-covalent. All other proteins (MBP-HPV16 E6/E7 and MBP-HPV18 E7) were subjected to the same processes with similar results demonstrating the expected mass determined by mass spectrometry (data not shown).

### Selection and performance of anti-HPV reference sera

The suitability of 6 individual donors, 3 HPV+ cervical cancer donors, and 6 monkey serum samples were evaluated as possible serum sets as reference serum in the MSD HPV multiplex immunoassay. Because the monkeys were vaccinated with the HPV antigens of interest, we hypothesized that an individual serum or pooled serum of non-human primates would provide the most robust serum sample set as a reference serum. In preliminary evaluation of the six monkeys, robust ECL signal was observed across all six samples screened at 3 different dilution factors (Fig 2A), and one monkey serum (Cyno 6470) was identified as a potential reference serum as this serum had the highest median titer against all HPV antigens compared to all animals and donors screened. Across the 11 dilutions ($100–1.04x10^8$) evaluated of potential monkey reference serum, three of the four HPV proteins showed marked signal above background to a dilution of 409,600-fold (Fig 2B) whereas HPV16 E6 lost linearity after the fifth dilution. Poor precision was also identified for anti-HPV16 E6 in monkey serum at a dilution factor of 1,600 where 4 of the 6 monkeys evaluated had intra-assay variance ranging from 37.8% to 89.5% against HPV16 E6 (Fig 2C) demonstrating that the vaccinated monkey serum would not be reliable in routine testing.

In evaluating human serum, we found broad reactivity to HPV antigens in the multiplex assay. On average, the HPV+ cervical cancer patients had elevated antibody signal against all HPV antigens tested compared to adult donors, however, HPV+ cervical cancer patients had elevated ECL signal to HPV16 E7 at a 1,000-fold dilution (Fig 2D). We identified two normal donor serum with elevated ECL signal against all antigens (BRH1516437 and BRH1516438)

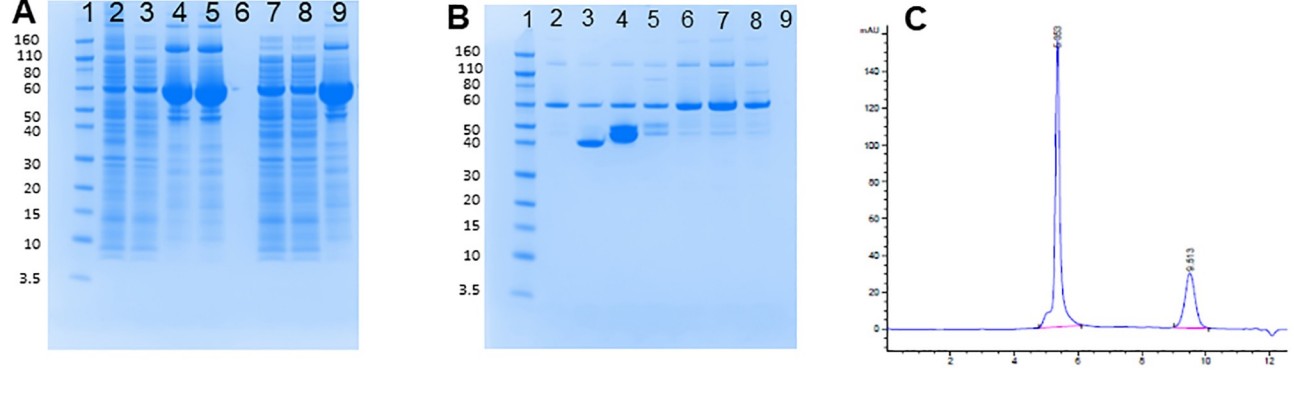

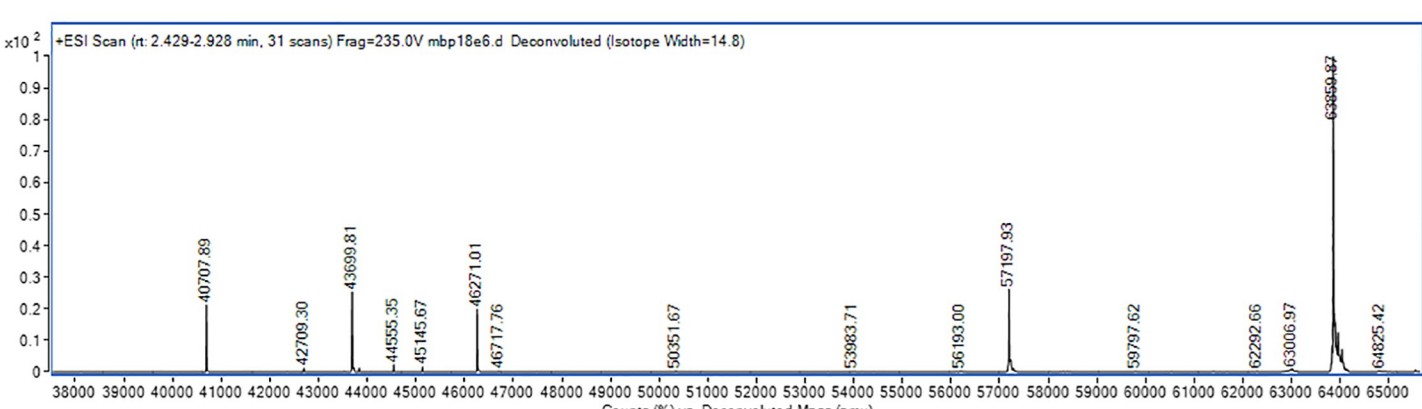

**Fig 1. Characterization of MBP-tagged HPV18 E6 protein via SDS-PAGE, size-exclusion chromatography, and mass spectrometry.** A–SDS-PAGE gel of MBP-trap purification of pMal MBP-18E6. Lane 1 designated the molecular weight marker, lane 2 is protein loaded on the column, lane 3 is the flow-through from the column, lanes 4 and 5 are independent collections of the peak of the primary elution, lane 6 is empty, lanes 7–9 is raw load protein (7), column flow-through (8), and peak of the primary elution (9) subjected to DTT reduction. B–SDS-PAGE gel of MBP-HPV18 E6 protein prior to and following DTT reduction with a Hi trap Q anion exchange column. Lane 1 designated the molecular weight marker, lane 2 is protein loaded on the column, lane 3 is the flow-through from the column, lanes 4 and 5 are independent collections of the peak of the primary elution, lanes 6–8 is raw load protein (6), column flow-through (7), and peak of the primary elution (8) subjected to DTT reduction. C–Size-exclusion chromatography of the peak of the primary elution subjected to DTT reduction in Fig 1A. D–Mass spectrometry of MBP-HPV18 E6 protein from re-folded and reduced, peak eluate solution from the QHP column.

and pooled these serums to create our reference serum. A standard curve was generated over 5 assay runs for the reference serum to assign concentrations in AU/mL for each analyte in the multiplex (Table 1). Reference serum intra-assay variance between duplicates ranged from 2.8% to 18.4% demonstrating reproducibility within experiments.

## Specificity of the MSD HPV multiplex assay

First, specificity of the multiplex immunoassay was evaluated by evaluating cross-reactivity with mouse monoclonal antigen-specific anti-HPV antibodies to HPV16/18 E6, HPV16 E7, HPV18 E6, and HPV18 E7 (n = 24 tests per antibody). Fig 3A demonstrates high affinity, mouse monoclonal antibodies produce signal as expected against target antigens with less than 3% cross-reactivity. As expected, high signal was observed for both HPV16 E6 and HPV18 E6 with the mouse monoclonal anti-HPV16/18 E6 antibody with a 2.6-fold increase in signal intensity for HPV18 E6 responses compared to HPV16 E6 at 0.25 μg/mL antibody concentration. Few specific monoclonal antibodies are commercially available for HPV16 E6, and those that are available have high variability and low ECL signal (data not shown).

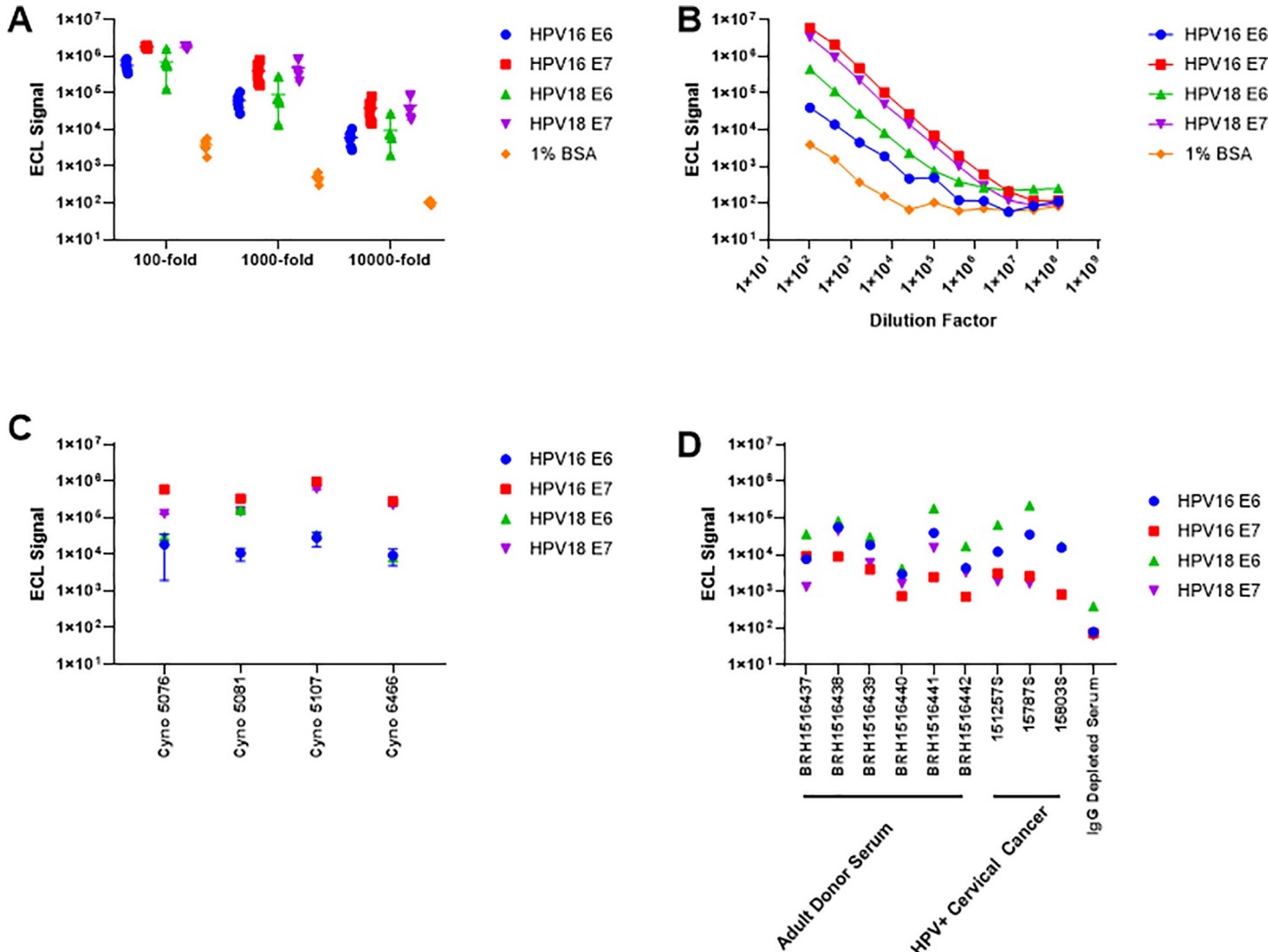

**Fig 2. Performance and selection of a panel of human and cynomolgus monkey serum in multiplex assay development.** A–Electrochemiluminescent (ECL) signal observed from the normal, adult healthy human donors (n = 6) for all antibodies tested in each well. Serum was evaluated at 3 different dilutions and plotted as the median ECL signal ± range. B–Concentration curves of observed ECL signal at 4-fold dilutions for Cyno 6470 for all HPV antibodies. C–Inter-assay variability observed for cynomolgus monkey serum (n = 4) for all HPV antibodies screened. D–intra-subject variability in antibody concentrations from normal, healthy adult donor serum, HPV+ cervical cancer subjects and IgG depleted serum to identify a reference serum sample. Samples BRH1516437 and BRH1516438 were selected and pooled as the reference material for subsequent assays.

Specificity of the assay was also evaluated by spiking soluble HPV16/18 E6 and E7 antigens at 10-fold greater mass than the amount coated in the well with pre-diluted antigen-specific antibodies, normal donor serum (n = 4), or HPV+ cervical cancer patient serum (n = 2). Expected depletion of signal was observed for antigen-specific monoclonal antibodies (Fig 3B) except for anti-HPV16 E6 where we observed a 52% reduction in signal. Additionally, HPV16 E6 competition was less efficient in reducing the anti-HPV16 E6 signal in human serum samples (Fig 3C). For the six human serum samples evaluated, the range of percent inhibition of HPV16 E6 was 6%-77% with a median inhibition of 40.5% (S1 Table).

Interestingly, antibodies to HPV18 E6 are as effective at inhibiting HPV16 E6 signal as its own signal in the assay indicating modest cross-reactivity due to nearly 50% sequence homology between these two proteins [19, 20]. One serum sample had significant antibody cross-

**Table 1. Concentration assignment of the pooled human donor reference serum from observed ECL signal.**

| Dilution factor | HPV16 E6 | | HPV16 E7 | | HPV18 E6 | | HPV18 E7 | |
|---|---|---|---|---|---|---|---|---|
| | ECL Signal | Assigned Concentration (AU/mL) | ECL Signal | Assigned Concentration (AU/mL) | ECL Signal | Assigned Concentration (AU/mL) | ECL Signal | Assigned Concentration (AU/mL) |
| 100 | 402403 | 30 | 88255 | 5 | 623159 | 50 | 315200 | 20 |
| 400 | 107323 | 7.5 | 23485 | 1.25 | 185969 | 12.5 | 77691 | 5 |
| 1600 | 28967 | 1.875 | 4541 | 0.313 | 51975 | 3.125 | 20165 | 1.250 |
| 6400 | 7218 | 0.469 | 956 | 0.078 | 13054 | 0.781 | 4689 | 0.313 |
| 25600 | 1832 | 0.117 | 330 | 0.020 | 3378 | 0.195 | 1269 | 0.078 |
| 102400 | 588 | 0.029 | 152 | 0.005 | 1131 | 0.049 | 369 | 0.020 |
| 409600 | 179 | 0.007 | 92 | 0.001 | 510 | 0.012 | 148 | 0.005 |
| Blank | 62 | | 91 | | 316 | | 54 | |

reactivity with all antigens being analyzed less than 40% of maximal signal (range = 1–40%). This serum may have reactivity to common epitopes or to the maltose binding protein (MBP) portion of the antibody tag [19]. Additionally, 5 of 6 of the serum samples evaluated inhibited to the same magnitude for incubation with HPV18 E6 antigen and detecting HPV16 E6 antibodies demonstrating HPV16 and 18 E6 may be equally detected. Robust signal depletion was seen for both E7 proteins in all tested human serum samples. Three HPV+ cervical cancer patient samples were incubated with saturating concentrations of MBP (20 µg/mL) to examine whether we were detecting any anti-MBP antibodies thereby increasing background signal. Fig 3D demonstrates that the detection of anti-HPV16 E7 antibodies by ELISA was unencumbered across all dilutions of serum tested following sample incubation with or without MBP. Given these results, the assay can robustly detect type-specific HPV antibodies, however low affinity, polyclonal antibodies to both E6 targets may be difficult in differentiating.

## Validation of the MSD HPV multiplex assay

The precision of the HPV multiplex immunoassay was evaluated based on intra-and inter-plate concentration assignments using human serum. 6 assay runs were completed evaluating 17 human serum donors at 500-fold dilution: 3 HPV+ cervical cancer patient serum (Group A), 4 human adult donor serum (Group B), and 10 pediatric donor serum (Group C) and IgG antibody depleted human serum was used as a negative control (Fig 4A–4D). In sample analysis, a broad range of reported concentrations was observed (21.29–13,534.79 AU/mL) with inter-run average %CV between 4.8%-23.9%, within typically established 75–125% of variance for ECL-based assays (S2 Table). Furthermore, the average concentration of intra-plate %CV of duplicates for all samples against all HPV antigens tested ranged from 1.5%-13.9% indicating that minimal ECL signal differences are observed when assayed in duplicate. From this analysis, we found that the reportable median concentrations 3 of 4 antigens (HPV16 E6, HPV18 E6 and HPV18 E7) were greater in adult and HPV+ cervical cancer serum samples compared to pediatric samples (Fig 4E). The median concentrations of the HPV+ cervical cancer group was 1.5-fold and 2.2-fold higher for HPV 16 E6 and HPV18 E6, respectively compared to adult donor serum. We expected the concentrations of pediatric samples to be approximately 0 units/mL however we found that there was a concentration range of 21–1881 units/mLwhich may be attributed to maternal antibodies generated *in utero* [21]. The median concentration of HPV16 E7 antibodies was no different between pediatric compared to adult donors as the average signal was 4-5-fold above the IgG depleted serum ECL levels.

Dilutional linearity of the assay was evaluated using 7 human serum samples and the reference serum. Intra-assay quadruplicates were evaluated for reactivity to all antigens between

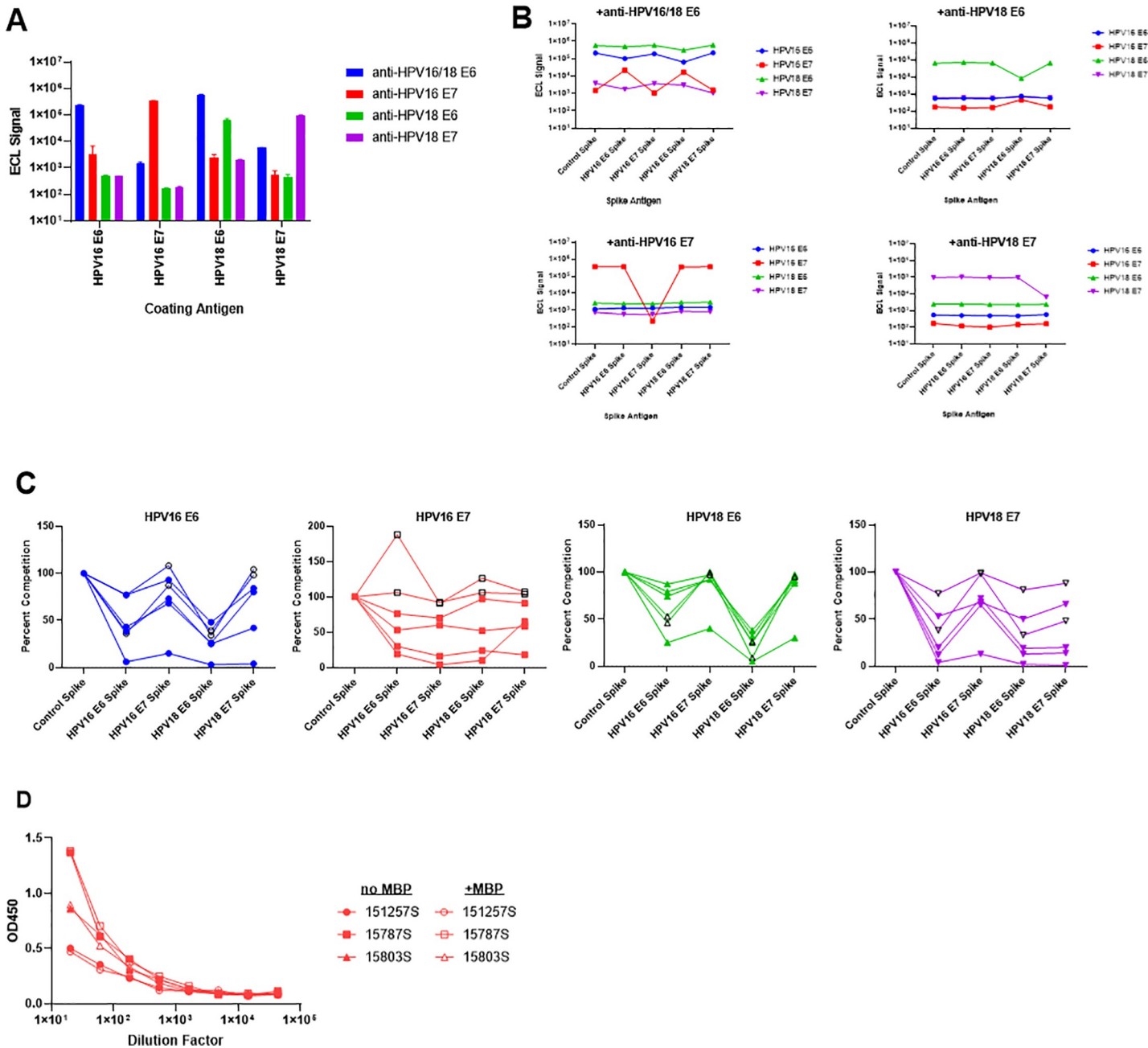

**Fig 3. Specificity of the multiplex HPV serology assay.** A–Specificity of mouse monoclonal anti-HPV antibodies to the coating antigen in each well as a total average ECL signal of 24 wells tested ± S.D. B–Competition assay to compete antigen and matched antibody signal. Monoclonal anti-HPV antibodies were incubated with the antigen and assayed to evaluate reduction of antibody-specific ECL signal. C–Competition assay on normal, healthy adult donor serum samples (n = 4; filled shapes) and HPV+ cervical cancer subjects (n = 2, black outline, open shapes). Serum samples were incubated with the antigen and assayed for percent inhibition of signal. D–Competition ELISA evaluating the contribution of anti-MBP antibodies in HPV+ cervical cancer subjects (n = 3). 10 μg of MBP was added to subject serum and levels of HPV16 E7 signal ($OD_{450}$) are plotted as a function of dilution factor of serum.

dilutions of 100–12,800-fold as shown in Fig 5A–5D. Normalizing to the 800-fold dilution, we found that a majority of the samples reported linear concentrations (±30%) between 400–6,400-fold dilution. These results confirm that sample testing at two dilutions: 500-fold and 10,000-fold are reasonable to determine HPV antibody concentration. In evaluating the

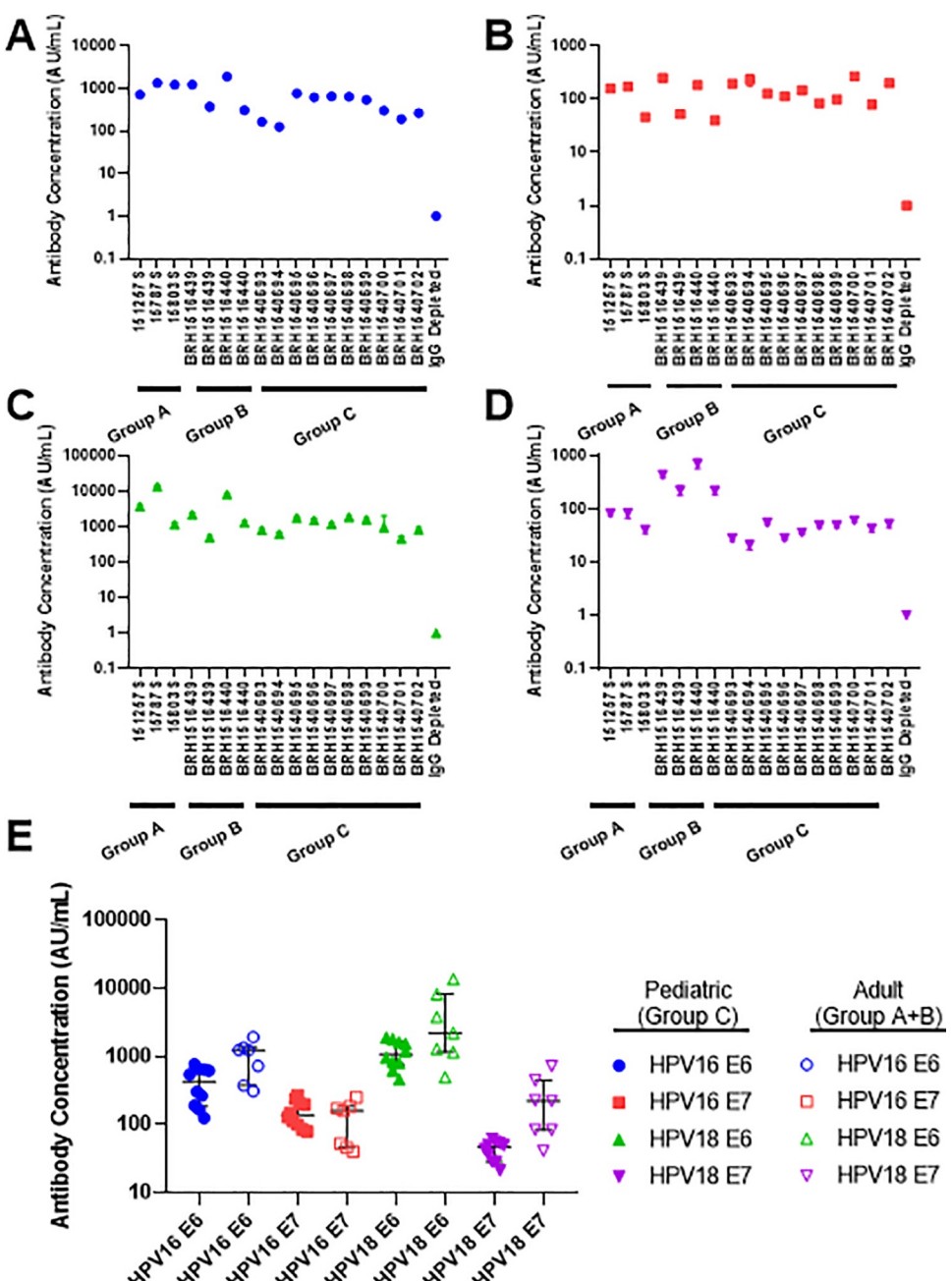

**Fig 4. Reproducibility and validation of the HPV multiplex MSD-based serology assay for various donor groups.**
Group A–HPV+ cervical cancer subjects (n = 3); Group B–normal, healthy adult donors (n = 4); Group C–normal
pediatric donors (n = 10). A–inter-subject variability for evaluating the concentration of IgG anti-HPV16 E6 over
multiple assays (n = 5 assays). B–inter-subject variability for evaluating the concentration of IgG anti-HPV16 E7 over
multiple assays (n = 6 assays). C—inter-subject variability for evaluating the concentration of IgG anti-HPV18 E6 over
multiple assays (n = 5 assays). D—inter-subject variability for evaluating the concentration of IgG anti-HPV18 E7 over
multiple assays (n = 5 assays). E–Inter-group anti-HPV IgG concentrations plotted as median ± interquartile range.
Open shapes are Groups A and B and filled shapes is Group C.

linearity of the reference serum, serum was diluted 4-fold across 11 concentrations starting at
a 100-fold dilution. Fig 5E demonstrates that linearity of the reference serum is linear from
100-fold to $10^6$-fold for all HPV antibodies, however, the recovery, due to diluting of the

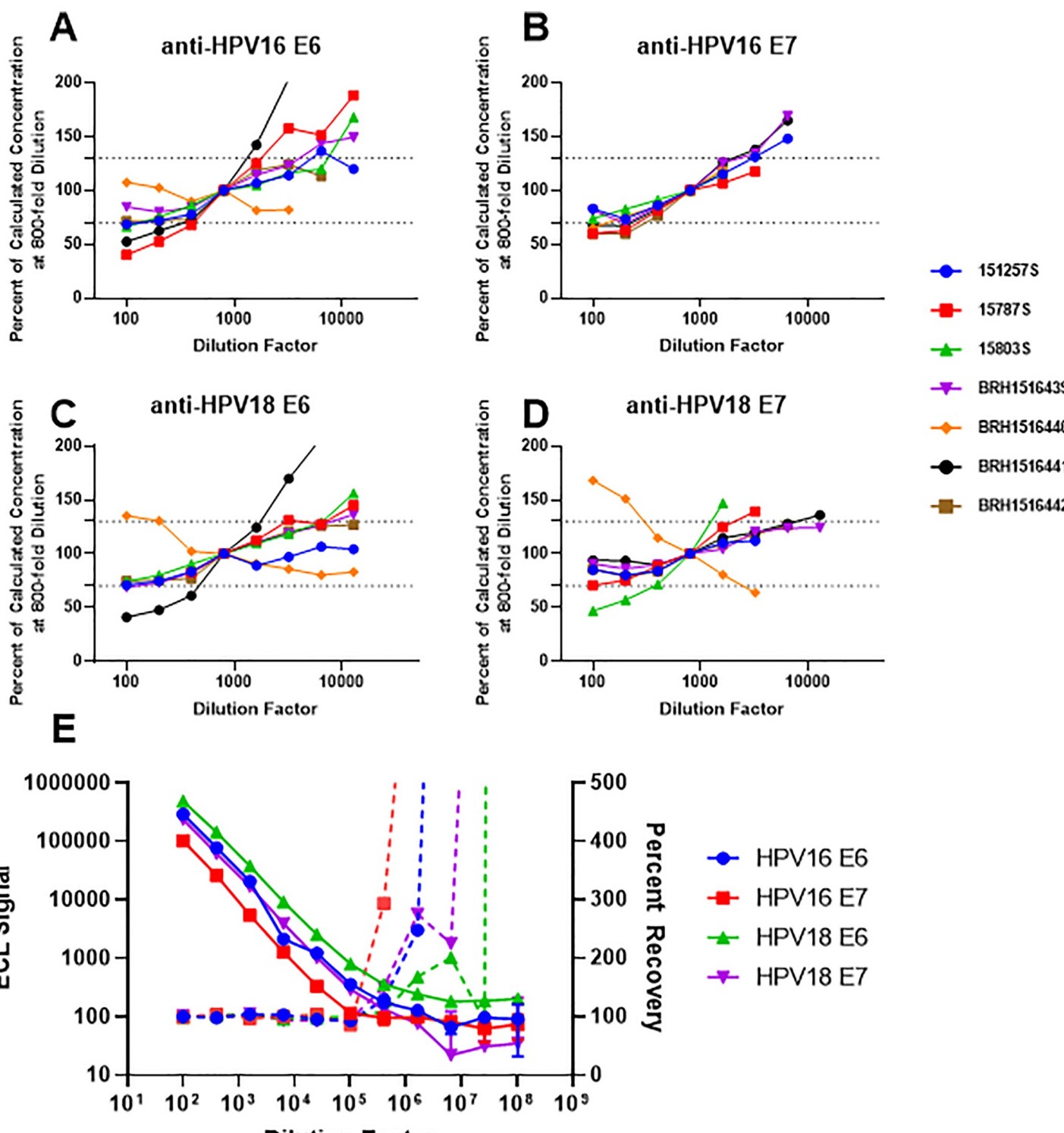

**Fig 5. Sample-dependent and reference material dilutional linearity in the multiplex anti-HPV MSD assay.** Serum from HPV+ cervical cancer subjects (n = 3), and normal, healthy adult donors (n = 4) was evaluated to identify the linear range for sample testing over 2 assays. Concentration for each sample was normalized to the observed concentration at 800-fold and plotted as a percent change from this value across 8 dilutions. A–dilutional linearity for seven samples in relation to the reference standard for anti-HPV16 E6 concentration. B–dilutional linearity for seven samples in relation to the reference standard for anti-HPV16 E7 concentration. C–dilutional linearity for seven samples in relation to the reference standard for anti-HPV18 E6 concentration. D–dilutional linearity for seven samples in relation to the reference standard for anti-HPV18 E7 concentration. E–Dilutional linearity of the reference serum over three assays runs plotted as ECL signal (left axis; solid lines) or percent recovery (right axis; dashed lines) ± %CV as a function of dilution factor of serum.

**Table 2. Estimated assay upper limit of quantitation (ULOQ) and percent recovery for type-specific anti-HPV concentrations from reference serum.**

| Antigen | Expected Concentration (AU/mL) | Observed Concentration (AU/mL) | Percent Recovery | Intra-plate % CV | Inter-plate % CV | Estimated ULOQ (AU/mL) |
|---|---|---|---|---|---|---|
| HPV16 E6 | 30 | 30.8 | 103% | 4.3 | 6.2 | 25.5 |
| | 25 | 26.5 | 106% | 3.4 | 5.8 | |
| | 20.8 | 21.9 | 105% | 1.8 | 3.8 | |
| | 17.4 | 18.5 | 106% | 3.8 | 7.4 | |
| HPV16 E7 | 5 | 5.3 | 106% | 4 | 8.7 | 4.25 |
| | 4.2 | 4.5 | 107% | 3.5 | 6.6 | |
| | 3.5 | 3.8 | 109% | 1.4 | 9 | |
| | 2.9 | 3.2 | 110% | 6.5 | 13.1 | |
| HPV18 E6 | 50 | 48.9 | 98% | 4 | 6.3 | 42.5 |
| | 41.7 | 43.4 | 104% | 4.8 | 6.4 | |
| | 34.7 | 36.6 | 105% | 3.5 | 4 | |
| | 28.9 | 30.2 | 104% | 5.4 | 7.4 | |
| HPV18 E7 | 20 | 19.8 | 99% | 6.3 | 6.3 | 17 |
| | 16.7 | 16.9 | 101% | 4.4 | 5.3 | |
| | 13.9 | 14.1 | 101% | 4.4 | 4.8 | |
| | 11.6 | 11.3 | 97% | 10.4 | 12.1 | |

serum, is lost for all antibodies at $10^5$-fold dilution. From this data, we could reliably calculate antibody concentration to HPV antigens between 100 and $10^5$-fold from the reference curve.

Next, we evaluated the upper (ULOQ) and lower limits of quantitation (LLOQ) of the assay. The ULOQ material was prepared by spiking the reference serum into assay diluent, and 4 dilutions at 1.2-fold were evaluated for percent recovery and intra- and inter-plate variability. Table 2 demonstrates that the recovery of the ULOQ material was within the expected 80–120% across all HPV antibodies. The intra- and inter-plate %CV were less than 20% and 25%, respectively indicating robust detection at the upper limit of the assay. Given this data, we set the ULOQ of the assay at 85% of the top tested concentration of antibody in the well for all analytes.

To establish the LLOQ of the assay, we created two series of samples by creating the first series with a pre-dilution of 500-fold and diluting serum in 1.5-fold steps across 8 dilutions. The second series was pre-diluted 500-fold with an additional 5-fold dilution and evaluated across 8 dilutions in 1.5-fold steps. We found that our assay lost sensitivity by the third dilution from all antigens on LLOQ series evaluated as indicated by the increase in inter-plate %CV above 25%. Our assay was able to detect to as low as 600 ECL units for anti-HPV16 E7 and as low as 2,500 ECL units for anti-HPV18 E6. These signals from the reference material result in the LLOQ assay ranging from 0.0457–0.1265 AU/mL (Table 3).

Next, we evaluated the robustness of the assay across a range of serum sample types. We collected 30 serum samples from adults with one or less sexual partner as a suspected low antibody concentration serum set and compared this to our panel of normal adult donors (n = 6) in combination with the HPV+ cervical cancer patient samples (n = 3) as a likely high antibody concentration subset. Fig 6A shows the range and median concentration of antibody observed for each donor cohort. The median concentration in the high group ranged between 19.2-fold to 46.1-fold greater than the low group. The greatest increase was in the anti-HPV18 E6 group whereas the lowest increase was observed in the anti-HPV18 E7 group. These results demonstrate that our assay can differentiate between people with high and low antibody titers and can stratify patients based on their pre-existing antibody concentrations.

Lastly, we selected 2 normal healthy donor serum (BRH1516441, BRH1516442) and 2 HPV + cervical cancer donor serum (15787S, 15803S) and evaluated sample stability at five

**Table 3. Estimated assay lower limit of quantitation and percent recovery for type-specific anti-HPV concentrations of the reference serum.**

| Antigen | Series 1 | | | | | Series 2 | | | | | Estimated LLOQ | |
| --- | --- | --- | --- | --- | --- | --- | --- | --- | --- | --- | --- | --- |
| | Expected Concentration (AU/mL) | Observed Concentration (AU/mL) | Percent Recovery | Intra-plate %CV | Inter-plate %CV | Expected Concentration (AU/mL) | Observed Concentration (AU/mL) | Percent Recovery | Intra-plate %CV | Inter-plate %CV | Signal | Concentration (AU/mL) |
| HPV16 E6 | 0.600 | 0.6953 | 116% | 7.1 | 16.8 | 0.120 | 0.0973 | 81% | 4.3 | 15.3 | 1000 | 0.0681 |
| | 0.400 | 0.4141 | 104% | 4.8 | 16.8 | 0.080 | 0.0729 | 91% | 3.7 | 11.3 | | |
| | 0.267 | 0.2765 | 104% | 3.2 | 12.9 | 0.053 | 0.0509 | 95% | 2.7 | 19.2 | | |
| | 0.178 | 0.1623 | 91% | 4.9 | 25.9 | 0.036 | 0.0305 | 86% | 3.6 | 22.6 | | |
| | 0.119 | 0.1112 | 94% | 7.3 | 19.6 | 0.024 | 0.0187 | 79% | 16.0 | 36.1 | | |
| | 0.079 | 0.0762 | 96% | 5.0 | 30.5 | 0.016 | 0.0131 | 83% | 21.6 | 33.7 | | |
| | 0.053 | 0.0474 | 90% | 11.5 | 36.8 | 0.011 | 0.0087 | 83% | 8.0 | 30.1 | | |
| | 0.035 | 0.0297 | 85% | 3.3 | 32.6 | 0.007 | 0.0051 | 73% | 31.1 | 46.7 | | |
| HPV16 E7 | 0.100 | 0.1182 | 118% | 7.0 | 11.4 | 0.020 | 0.0177 | 89% | 22.9 | 24.0 | 600 | 0.0457 |
| | 0.067 | 0.0672 | 101% | 4.2 | 18.5 | 0.013 | 0.0135 | 101% | 24.4 | 24.6 | | |
| | 0.044 | 0.0443 | 100% | 7.6 | 18.3 | 0.009 | 0.0095 | 107% | 40.1 | 43.4 | | |
| | 0.030 | 0.0266 | 90% | 9.8 | 35.1 | 0.006 | 0.0059 | 100% | 42.7 | 44.6 | | |
| | 0.020 | 0.0201 | 102% | 11.0 | 19.7 | 0.004 | NE | NE | 56.8 | 56.8 | | |
| | 0.013 | 0.0143 | 109% | 15.6 | 35.6 | 0.003 | NE | NE | 64.8 | 64.8 | | |
| | 0.009 | 0.0074 | 84% | 22.5 | 36.8 | 0.002 | NE | NE | 39.9 | 42.8 | | |
| | 0.006 | 0.0047 | 80% | NE | 38.6 | 0.001 | NE | NE | 69.3 | 67.2 | | |
| HPV18 E6 | 1.000 | 1.1673 | 117% | 7.6 | 16.6 | 0.200 | 0.1511 | 76% | 20.0 | 12.5 | 2500 | 0.1265 |
| | 0.667 | 0.6814 | 102% | 5.5 | 18.8 | 0.133 | 0.1145 | 86% | 11.7 | 11.0 | | |
| | 0.444 | 0.4515 | 102% | 2.8 | 15.0 | 0.089 | 0.0813 | 91% | 18.9 | 17.4 | | |
| | 0.296 | 0.2649 | 89% | 4.7 | 29.5 | 0.059 | 0.0471 | 79% | 18.2 | 19.5 | | |
| | 0.198 | 0.1912 | 97% | 5.9 | 25.0 | 0.040 | 0.0301 | 76% | 24.0 | 26.1 | | |
| | 0.132 | 0.1281 | 97% | 7.1 | 35.5 | 0.026 | 0.0221 | 84% | 33.6 | 36.8 | | |
| | 0.088 | 0.0794 | 90% | 9.0 | 36.0 | 0.018 | 0.0144 | 82% | 37.7 | 24.0 | | |
| | 0.059 | 0.0507 | 87% | 5.7 | 32.6 | 0.012 | 0.0087 | 74% | 32.2 | 33.5 | | |
| HPV18 E7 | 0.400 | 0.4964 | 124% | 10.3 | 12.4 | 0.080 | 0.0642 | 80% | 21.6 | 15.9 | 1000 | 0.0561 |
| | 0.267 | 0.2732 | 102% | 5.4 | 13.0 | 0.053 | 0.0478 | 90% | 17.2 | 18.6 | | |
| | 0.178 | 0.1818 | 102% | 2.7 | 12.0 | 0.036 | 0.0327 | 92% | 19.5 | 20.0 | | |
| | 0.119 | 0.1056 | 89% | 7.5 | 27.4 | 0.024 | 0.0174 | 73% | 35.2 | 37.9 | | |
| | 0.079 | 0.0782 | 99% | 5.5 | 19.4 | 0.016 | 0.0125 | 79% | 40.4 | 42.1 | | |
| | 0.053 | 0.0494 | 94% | 5.7 | 31.4 | 0.011 | 0.0082 | 78% | 41.0 | 42.6 | | |
| | 0.035 | 0.031 | 88% | 8.9 | 32.0 | 0.007 | 0.0055 | 78% | 48.7 | 51.4 | | |
| | 0.023 | 0.0197 | 84% | 7.1 | 25.5 | 0.005 | 0.0033 | 70% | 40.3 | 33.5 | | |

conditions: overnight at 4˚C, overnight at room temperature, and through 3 freeze/thaw cycles at -80˚C. We observed modest reductions in observed antibody concentration when incubating samples overnight at room temperature (25–30%) for sample BRH1516442 whereas the other samples had a 20% reduction in concentration (Fig 6B). 10–25% reduction in antibody concentration was also seen at the third freeze/thaw cycle. These freeze/thaw results confirm the suitability of the assay for HPV antibody concentration assignment.

## Comparison of anti-HPV16 E7 signal in MSD multiplex immunoassay to ELISA

In order to establish suitability of the assay to standard methods, we evaluated 26 serum samples (16 healthy donor, 10 HPV+ cervical cancer) in a head-to-head assay across 8 serial

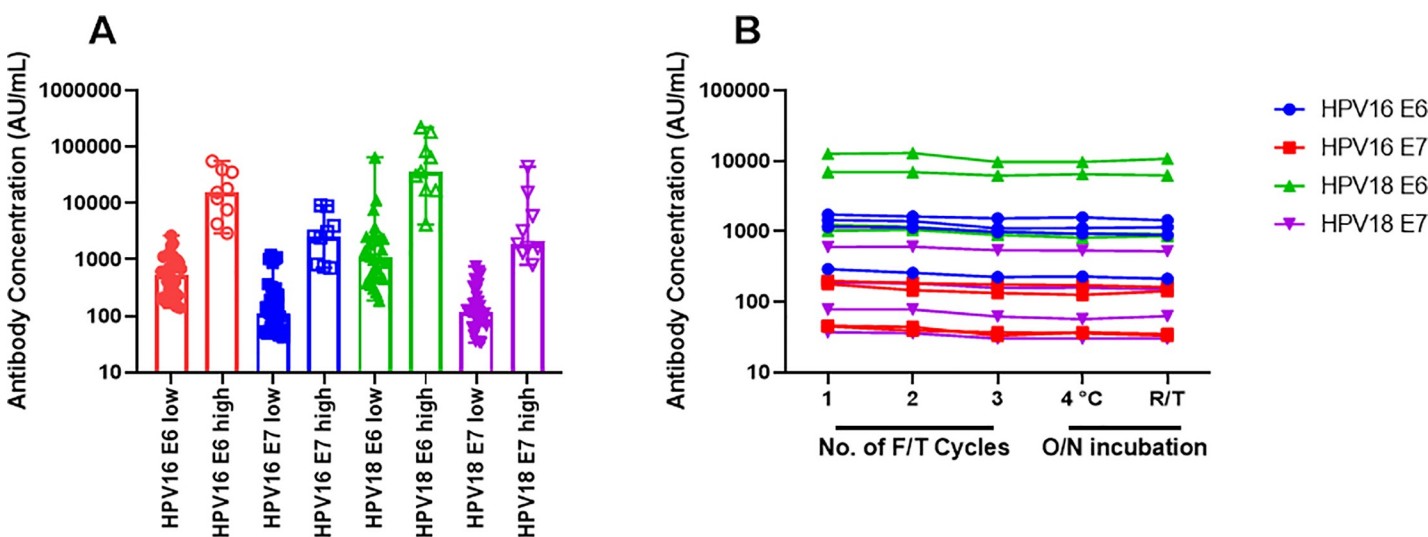

**Fig 6. Multiplex anti-HPV immunoassay robustness.** A–Suspected anti-HPV antibody low subjects (n = 30) versus suspected high anti-HPV subjects (n = 9) including previously assayed normal, healthy adult donor serum (n = 6) and HPV+ cervical cancer patient serum (n = 3) is plotted as median antibody concentration ± range. B– Effect of freeze-thaw cycles (1–3 freeze-thaw cycles), overnight (O/N) incubation at 4°C or room temperature (R/T) on anti-HPV concentrations for 4 samples: BRH1516439, BRH1516440, 15787S, and 15803S.

dilutions resulting in 208 evaluable data points between ELISA and multiplex MSD for detecting anti-HPV16 E7 responses. We analyzed the data using Spearman's Rank-Order Correlation and Pearson's Correlation as shown in Fig 7. The Spearman's coefficient was calculated at r = 0.7775 as the data are logarithmic with respect to ECL signal from MSD and linear for $OD_{450}$ values from ELISA. Given the positive value and r value of Spearman's correlation approaching 1 indicates that the results are in strong agreement and that the technologies are

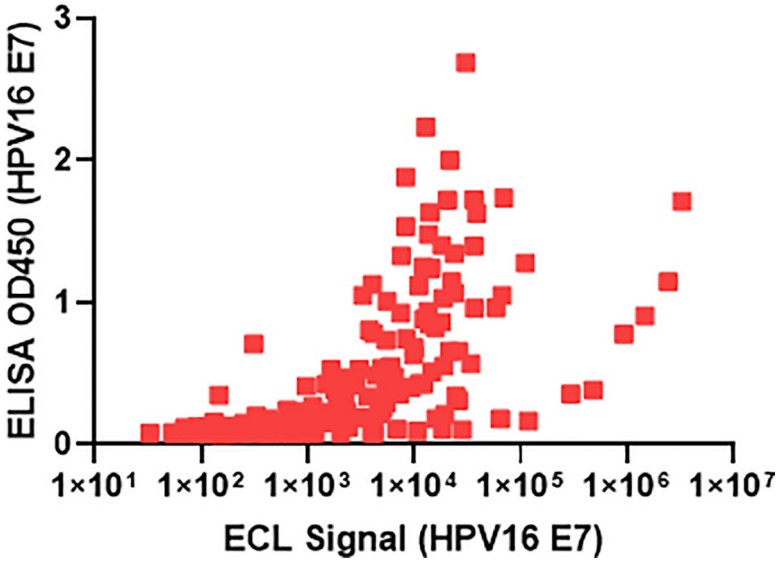

**Fig 7. Correlation between anti-HPV16 E7 responses measured by ELISA and MSD multiplex.** 208 evaluable data points were plotted from a head-to-head assay with 26 matched donor samples (16 normal, healthy adult donor, 10 HPV+ cervical cancer) across 8 serial dilutions for anti-HPV16 E7 signals. Log axis for ECL signal is plotted on the x-axis and linear axis for $OD_{450}$ ELISA values are plotted on the y-axis to evaluate the distribution and Spearman's Coefficient.

comparable. Additionally, when log-transforming the data, a Pearson coefficient can be used to describe the similarity of the data. The Pearson's correlation coefficient calculated (*r*) is *r* = 0.6692. While this *r* value is below a standard cut-off 0.7, the data is moderately linear. The data also demonstrate the range of signal detected by the MSD assay as it spans at least 5-logs of signal whereas the ELISA assay affords ~3-logs of signal.

## Discussion

This report describes a multiplex assay leveraging the spot printing of the MSD electrochemi-luminescent technology based on the fusion tagging of HPV oncoproteins to simultaneously detect antibody concentrations to HPV16 E6 and E7 as well HPV18 E6 and E7 in a high-throughput manner with a small amount of sample needed to conduct the assay (500-fold dilution of sample in a 50 μL volume). The synthesis of these novel oncoproteins allowed us to detect a wide range of antibody concentrations to HPV proteins in human serum, HPV+ cervical cancer serum, and vaccinated subjects to HPV oncoproteins with precision, reproducibility, and some degree of cross-reactivity between HPV16 and 18 E6 antibodies.

Multiplex serology assays require less time and normalization efforts than traditional serology assays generally performed under standard ELISA methods. Here, our data generated for HPV16 E7 antibodies demonstrates excellent concordance, however MSD affords us two additional magnitudes of dynamic range. This range affords us the ability to detect low affinity polyclonal antibodies that may be detected approaching or below potential background in standard ELISA [20–23]. While we hypothesized that HPV+ subjects would have greater antibody concentrations, we found a modest antibody concentration increase in HPV+ subjects to normal healthy donors. This may be attributed to the broad prevalence of HPV16 and 18 positivity in the general population due to the number of sexual partners [24,25] lending itself the need to identify a true negative population to set serostatus cutoff values from individuals with less than 1 sexual partner. Furthermore, we found low antibody concentrations in our pediatric donor cohort which may be attributed to material antibodies [21] as these children were between ages 1–5 days. In setting a serostatus cutoff value for these HPV antibodies, we would have increased confidence in identifying pharmacodynamic attributes of patients on current HPV vaccine trials targeting HPV16 and 18 oncoproteins E6 and E7, however we cannot rule out that the observed signal in non-HPV+ populations may be attributed to some degree of a lack of specificity [6,10,11].

Interestingly, vaccinated monkey serum to HPV16/18 E6 and E7 demonstrated excellent inter-assay precision to all antibodies except anti-HPV16 E6. This phenomenon was not observed in human serum and could be attributed to differing reactive epitopes [20], or poor interaction between monkey and human tags in the detection antibody [26]. Furthermore, modest cross-reactivity was observed in most human serum screened for antibodies to both HPV16 and 18 E6. This could be attributed to similar sequence homology (about 50%) and broad polyclonal responses [19]. Importantly, this effect was not observed for all 6 donors evaluated and is not attributed to the MBP fusion tag showing that the similar sequences and polyclonal nature of HPV antibodies may lend itself to modest cross-reactivity. Follow-up genotype testing on these subjects to identify HPV16 versus HPV18 positive individuals would be critical to understand which antigen or antibody a subject may have reactivity to as well as enhancing the specificity of our assay.

HPV E6 oncoproteins are difficult to synthesize due to the nature of their aggregation and lack of solubility due to cysteine-rich domains [14]. To overcome this, our E6 and E7 oncoproteins were synthesized with a maltose binding protein. Other groups have evaluated crystal structures of the p53 core domain binding pocket of HPV16 E6 when bound to MBP [8] and

demonstrated better data reproducibility compared to other protein tags such as His, myc, or tag fusion proteins owing to the hypothesis that MBP-tagged E6 and E7 oncoproteis may be superior in quality over alternatively tagged recombinant proteins. MBP-tagged proteins can also be used as affinity tags on various resins [27] and here we demonstrate strong binding, remarkable recovery, moderate aggregation, and enhanced solubility of difficult proteins to synthesize due to the ability to capture and fold proteins on-column. The E6 and E7 oncoprotein reagents developed here demonstrate utility in an assay to evaluate the concentration of type-specific anti-HPV in serum of normal and cancer patients alike.

Our assay is the first assay explored and developed in a multiplex fashion in evaluating the serostatus of the key drivers of HPV+ oncogenesis in cervical and head and neck malignancies [2]. Other assays have explored the utility of antibodies to these oncoproteins in singlicate by ELISA [22], leveraged different technologies such as Luminex [17], other non-key drivers related to HPV oncogenesis such as HPV6, 11, 35 [28], or conjugation to pseudovirion particles [15,28–30]. Other platforms such as ELISA and Luminex do not afford the dynamic range of detection that MSD provides, nor do they maintain confirmation of the HPV16 and 18 proteins for robust analysis of serum samples [7,8]. In using our reference serum, we can reproducibly determine antibody concentrations for subjects in support of ongoing clinical trials in the HPV cervical and head and neck cancer space [6,9–12] or other indications linked to HPV positivity. Our results demonstrate reproducibility across HPV16/18 E6 and E7 antibodies for our pooled serum to directly calculate serum antibody concentrations.

The primary limitation to our assay is two-fold. First, linking of proteins to maltose binding protein may increase signal background due to presence of antibodies to MBP [31]. While these antibodies may be low in concentration, we cannot rule out inter-subject variability that may increase background, leading to artificially enhanced signal. Second, due to the polyclonal nature of HPV antibodies, we cannot rule out the cross-reactive nature of HPV16 and 18 E6 antibodies. It is currently unknown as to the quantity of cross-reactive antibodies that may exist between HPV16 and 18 E6 contributing to the possibility of increased, cross-reactive signal. A polyclonal HPV16 E6 or HPV18 E6 antibody would be critical in evaluating this potential phenomenon.

In conclusion, we developed a specific, sensitive, reproducible multiplexed serology assay for the simultaneous quantitation of IgG antibodies to HPV16/18 E6 and E7. Our assay shows strong correlation to pre-existing technologies and could be a good alternative to standard single-plex serology assays in large-scale epidemiology studies or clinical trials in evaluation of primary drivers in HPV-driven oncogenesis.

## Supporting information

**S1 Fig. Protein sequences of MBP-tagged proteins.** UniProt derived and MBP-tagged sequences of synthesized HPV16/18 E6 and E7 proteins.
(DOCX)

**S1 Table. Assay antibody-dependent competition.** Percent competition to target antibody following type-specific antigen spike with serum sample.
(XLSX)

**S2 Table. Samples evaluated for assay metrics.** Reproducibility and precision of samples evaluated to establish assay precision for anti-HPV16/18 E6 and E7.
(XLSX)

**S1 Raw images.**
(PDF)

## Acknowledgments

We thank Dr. Jean Boyer from Inovio Pharmaceuticals for her gift and supply of HPV16/18 E6 and E7 vaccinated cynomolgus monkey serum. We also thank Mark Billadeau, Samarth Chugh from MSD for input on coating optimization. Lastly, we thank Dr. Nathan Standifer for critical review of the manuscript toward expertise of serology-based assay development.

## Author Contributions

**Conceptualization:** Hans Layman.

**Data curation:** Hans Layman, Anastasia A. Aksyuk, Jill M. Dunty, Dusit Natrakul, Nithya Swaminathan, Christopher J. DelNagro.

**Formal analysis:** Hans Layman, Jill M. Dunty, Dusit Natrakul, Nithya Swaminathan, Christopher J. DelNagro.

**Investigation:** Hans Layman, Anastasia A. Aksyuk, Christopher J. DelNagro.

**Methodology:** Hans Layman, Keith W. Rickert, Susan Wilson, Anastasia A. Aksyuk, Dusit Natrakul, Nithya Swaminathan, Christopher J. DelNagro.

**Project administration:** Hans Layman, Jill M. Dunty.

**Resources:** Keith W. Rickert, Susan Wilson, Anastasia A. Aksyuk, Jill M. Dunty, Christopher J. DelNagro.

**Supervision:** Hans Layman, Christopher J. DelNagro.

**Validation:** Hans Layman, Anastasia A. Aksyuk.

**Visualization:** Hans Layman.

**Writing – original draft:** Hans Layman, Keith W. Rickert, Susan Wilson, Anastasia A. Aksyuk, Christopher J. DelNagro.

**Writing – review & editing:** Hans Layman.

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
