## [Decision Letter · Decision Letter 0]

22 Jan 2020

PONE-D-19-34611

Development and validation of a multiplex immunoassay for the simultaneous quantification of type-specific IgG antibodies to E6/E7 oncoproteins of HPV16 and HPV18

PLOS ONE

Dear Dr Layman,

Thank you for submitting your manuscript to PLOS ONE. After careful consideration, we feel that it has merit but does not fully meet PLOS ONE’s publication criteria as it currently stands. Therefore, we invite you to submit a revised version of the manuscript that addresses the points raised during the review process.

We would appreciate receiving your revised manuscript by Mar 07 2020 11:59PM. To enhance the reproducibility of your results, we recommend that if applicable you deposit your laboratory protocols in protocols.io, where a protocol can be assigned its own identifier (DOI) such that it can be cited independently in the future. For instructions see: http://journals.plos.org/plosone/s/submission-guidelines#loc-laboratory-protocols

We look forward to receiving your revised manuscript.

Kind regards,

Scott M. Langevin, Ph.D., M.H.A., C.T. (ASCP)

Academic Editor

PLOS ONE

2. Please clarify whether human (healthy donors, paediatric donors, cervical cancer patients) and animal (monkey) specimens were all sourced from third parties commercial or not) and state name and location of those parties. Please also clarify whether third parties had obtained ethical approval.

3. Thank you for stating the following in the Financial Disclosure section: 'No'

We note that one or more of the authors are employed by commercial companies: Gilead Sciences, Inc., Exelixis, Inc., AstraZeneca plc and Meso Scale Diagnostics, LLC

Reviewers' comments:

Reviewer's Responses to Questions

**Comments to the Author**

1. Is the manuscript technically sound, and do the data support the conclusions?

Reviewer #1: Partly

Reviewer #2: Yes

2. Has the statistical analysis been performed appropriately and rigorously? 

Reviewer #1: Yes

Reviewer #2: I Don't Know

3. Have the authors made all data underlying the findings in their manuscript fully available?

Reviewer #1: Yes

Reviewer #2: Yes

4. Is the manuscript presented in an intelligible fashion and written in standard English?

Reviewer #1: Yes

Reviewer #2: Yes

5. Review Comments to the Author

Reviewer #1: This manuscript reports on the development and initial evaluation of a multiplex assay to detect antibodies against the E6 and E7 proteins of human papillomavirus types 16 and 18. Extensive development and characterization was performed and is well described in the manuscript. However, the following issues identified by the authors could be addressed more explicitly in the Abstract and Discussion:

1. Several lines of evidence suggest presence of cross-reactivity in the assay. Given this, not only does it appear that “… we cannot rule out the cross-reactive nature of HPV16 and HPV18 E6 antibodies.” But, one might “rule in” the likely presence of such an effect. The Abstract and Discussion should be more upfront about this fact.

2. No mention is made in the Abstract and Discussion regarding the high detection of antibodies in pediatric (presumably unexposed) samples, which is suggestive of some non-specificity of the assay (Figure 4). In fact, average concentrations between 100 – 1,000 were noted for all antigens except for HPV-18 E7. This high detection of signal among individuals expected to be unexposed, together with the increased dynamic range of the assay, suggest that the authors’ attempt to increase the dynamic range (decrease the LLOQ) might have led to reduced specificity of the assay.

3. The final sentence of the Abstract is a bit of an overstatement since comparison with ELISA was performed for only one of the four antigens included in the multiplex assay.

4. The 4th paragraph of the Discussion seems speculative and is not based on data reported in the Results. This reviewer suggests dropping that paragraph, which makes statements about the advantage of the newly developed assay vis-a vis other existent platforms (including multiplex platforms) without providing any data to support such asserions.

Reviewer #2: The authors developed a specific, sensitive, reproducible multiplexed serology assay for the simultaneous quantitation of IgG antibodies to HPV16/18 E6 and E7 using an electrochemiluminescent technology. The data are convincing. However, although this assay seems to be a good alternative to standard ELISA, the authors should better explain how this technology may be used as an alternative to the Luminex technology. Bead-based assays are widely used and demonstred high performance in large-scale epidemiology studies (Waterboer et al., PMID: 16099939). Why the authors didn’t compare the performance of their assay with a bead-based assay (or serum previously characterized using a bead-based assay)? WHat about the cost for each sample? Also the authors should also explain better this technology on its way to detect specifically each of the antigens. It is not clear.

Lines 434-435; The authors found similar antibody concentrations in HPV+ subjects to normal healthy donors which is surprising as only antibodies to L1 should be found. Please explain. The E6 antibodies should be rare in the general population.

6. PLOS authors have the option to publish the peer review history of their article (what does this mean?). If published, this will include your full peer review and any attached files.

Reviewer #1: Yes: Allan Hildesheim

Reviewer #2: No

---

## [Author Response · Author response to Decision Letter 0]

30 Jan 2020

Rebuttal Letter

We appreciate the thorough review from Reviewers 1 and 2 and their comments regarding the quality of the data and to more clearly state our abstract and conclusions in the discussion. Below, we capture the comments from each reviewer and respond with the following edits that can be found in the marked-up text document.

Editor Comments:

Style requirements for file naming – addressed

Clarify whether human and monkey specimens were obtained from third parties commercial or not. Human samples were obtained from BioIVT (Hicksville NY) or Proteogenex (Inglewood, CA) with informed consent. This has been updated in the cover letter. The monkey serum samples were obtained as a gift from Jean Boyer and the animal protocol was approved with IACUC Board approval at Inovio Pharmaceuticals in San Diego, CA.

Funding statements and competing interests have been updated for all authors in the cover letter.

Reviewer 1:

1. Several lines of evidence suggest presence of cross-reactivity in the assay. Given this, not only does it appear that “… we cannot rule out the cross-reactive nature of HPV18 and HPV18 E6 antibodies.” But, one might “rule in” the likely presence of such an effect. The abstract and discussion should be more upfront about this fact.

We address the cross-reactivity effect upfront in the abstract more clearly (line 32 and 33) by stating that HPV18E6 is as good as neutralizing HPV16 E6 in 5 of 6 samples tested in the cross-reactivity experiment. Before it merely stated that cross-reactivity existed and was observed. Lines 288-290 now address this observation in the results. However, for the discussion, this is addressed in lines 480-485. We feel that the development of the assay and the metrics for qualification against a reference serum is most important, and that these observations could be found in sample testing to identify the reference serum. We did include a brief summary (lines 429-430) regarding the finding of the cross-reactivity up front in the discussion.

2. No mention is made in the abstract and discussion regarding the high detection of antibodies in pediatric (presumably unexposed samples), which is suggestive of some non-specificity of the assay (Figure 4). In fact, average concentrations, between 100-1,000 were noted for all antigens except for HPV18 E7. This high detection of signal among individuals expected to be unexposed, together with the increased dynamic range of the assay, suggest that the authors’ attempt to increase dynamic range (decrease the LLOQ) might have led to reduced specificity of the assay. 

We acknowledge the levels of basal activity in what we would presume would be an unexposed group to HPV. We have addressed this by updating the methods to include the age of the diagnostic retains of pediatric samples (1-5 day old infants; line 98). Having this information is important as maternal antibodies would be detected at this age and they may be at low concentrations. We have updated the text in the results to reflect this finding as well (lines 312-314) and included a reference from a CVI paper Matys et al from 2012 (PMID 22518014). This was also addressed in the discussion in lines 444-446.

3. The final sentence of the abstract is a bit of an overstatement since comparison with ELISA was performed for only one of the four antigens included in the assay.

We have removed the language regarding standard ELISA methods (lines 35-39) to show what we have really developed: an assay that is high throughput, multiplexed, a good dynamic range, and sample input.

4. The fourth paragraph of the discussion seems speculative and is not based on data reported in the results. This reviewer suggests dropping that paragraph, which makes statements about the advantage of the newly developed assay vis-à-vis other existing platforms (including multiplex platforms) without providing any data to support such assertions.

We feel that this section is necessary to describe the tag and recovery portion of the protein. Many other groups have tried other tags to HPV such as His, myc, etc and their results have not been as reproducible. We acknowledge this section is somewhat speculative and would require additional chemistry analysis but would be beyond the scope of the current paper.

Reviewer #2: The authors developed a specific, sensitive, reproducible multiplex serology assay for the simultaneous quantitation of IgG antibodies to HPV16/18 E6 and E7. The data are convincing. However, although the assay seems to be a good alternative to standard ELISA, the authors should better explain how this technology may be used as an alternative to the Luminex technology. Bead-based assays are widely used and demonstrate high performance in large-scale epidemiology studies (PMID: 16099939). Why the authors didn’t compare the performance of their assay with a bead-based assay (or serum previously characterized using a bead-based assay)? What about the cost for each sample? Also, the authors should explain better this technology on its way to detect specifically each of the antigens. It is not clear.

Point 1 – Why did the authors not compare the performance of their assay with a bead-based assay?

A previous study has evaluated a large cohort comparison of serum samples to HPV 6, 11, 16 and 18 L1 between MSD and Luminex (PMID: 25554636). We acknowledge there is a gap in our reagents toward their performance in a bead based assay. However, EDC crosslinking of proteins to beads may result in untoward effects of the labile HPV proteins. Because the MSD assay affords the protein to bind to the plate in its native confirmation post-synthesis, no chemical modifications are done to perturb the chemical structure. Text was added to the introduction (lines 69,70 and 75,76) to clarify the choice of MSD over Luminex and why we elected to not pursue Luminex.

Point 2 – What about the cost for each sample?

The cost of the sample was omitted from the manuscript because it does not contribute to the strength of the paper. However Luminex and MSD are comparable in cost. Roughly speaking MSD and Luminx would be around $200 per sample. Luminex would tend to be a little higher due to conjugation chemistry for proteins/antigens. 32 samples could be run per plate (in duplicate) with a standard curve in triplicate. For ELISA, sample cost would be significantly less, about $75/sample, however without the ability to multiplex, the cost per sample increases to $300+ and a much larger volume is needed. For precious samples, this could be a severe limitation.

Point 3 – The authors should better explain this technology on its way to detect specifically each of the antigens.

This is highlighted from point 1 in so far that the confirmation of the protein/antigen to the plate and its confirmation being conserved whereas with Luminex, conjugation chemistry may result in protein degradation.

Point 4 – Lines 434-435; the authors found similar antibody concentrations in HPV+ subjects to normal healthy donors which is surprising as only antibodies to L1 should be found. Please explain. The E6 antibodies should be rare in the general population.

We do agree that the antibody concentrations are similar, however this analysis needs to be expanded over a larger sample size to get true statistical differences and the scope of the manuscript is to report on the method development, and, secondarily, to identify who may be seropositive. We have adjusted the text to reflect the data shown in Fig 4. We did observe elevated median concentrations of E6 antibodies in patients with HPV+ infections (HPV16 and 18) of 1.5 – 2.2 fold (lines 317-320). We cannot rule out that the adult donor population may have an HPV infection not linked to cancer due to the source of our adult donor cohort which was unspecified for the number of sexual partners. It has been noted previously that women with 4+ sexual partners had seropositivity to HPV antibodies in >60% of serum evaluated (PMID: 11724843). We updated our conclusion (lines 447-453) to reflect the need to identify a truly negative population in additional assay development.

---

## [Decision Letter · Decision Letter 1]

6 Feb 2020

PONE-D-19-34611R1

Development and validation of a multiplex immunoassay for the simultaneous quantification of type-specific IgG antibodies to E6/E7 oncoproteins of HPV16 and HPV18

PLOS ONE

Dear Dr Layman,

Thank you for submitting your manuscript to PLOS ONE. After careful consideration, we feel that it has merit but does not fully meet PLOS ONE’s publication criteria as it currently stands. Therefore, we invite you to submit a revised version of the manuscript that addresses the points raised during the review process.

We would appreciate receiving your revised manuscript by Mar 22 2020 11:59PM. To enhance the reproducibility of your results, we recommend that if applicable you deposit your laboratory protocols in protocols.io, where a protocol can be assigned its own identifier (DOI) such that it can be cited independently in the future. For instructions see: http://journals.plos.org/plosone/s/submission-guidelines#loc-laboratory-protocols

We look forward to receiving your revised manuscript.

Kind regards,

Scott M. Langevin, Ph.D., M.H.A., C.T. (ASCP)

Academic Editor

PLOS ONE

Reviewers' comments:

Reviewer's Responses to Questions

**Comments to the Author**

1. If the authors have adequately addressed your comments raised in a previous round of review and you feel that this manuscript is now acceptable for publication, you may indicate that here to bypass the “Comments to the Author” section, enter your conflict of interest statement in the “Confidential to Editor” section, and submit your "Accept" recommendation.

Reviewer #1: (No Response)

Reviewer #2: All comments have been addressed

2. Is the manuscript technically sound, and do the data support the conclusions?

Reviewer #1: Yes

Reviewer #2: Yes

3. Has the statistical analysis been performed appropriately and rigorously? 

Reviewer #1: Yes

Reviewer #2: I Don't Know

4. Have the authors made all data underlying the findings in their manuscript fully available?

Reviewer #1: Yes

Reviewer #2: (No Response)

5. Is the manuscript presented in an intelligible fashion and written in standard English?

Reviewer #1: Yes

Reviewer #2: (No Response)

6. Review Comments to the Author

Reviewer #1: In reviewing the responses to original comments and the revised manuscript, I have only one residual comment as follows:

In addressing the higher than expected positivity observed among pediatric subjects, the authors are correct in stating that one explanation is the presence of maternal antibodies. However, they fail to explicitly state that an alternative explanation is lack of assay specificity (i.e., pediatric positives are false positives). This should be explicitly stated in the manuscript.

Reviewer #2: (No Response)

7. PLOS authors have the option to publish the peer review history of their article (what does this mean?). If published, this will include your full peer review and any attached files.

Reviewer #1: Yes: Allan Hildesheim

Reviewer #2: No

---

## [Author Response · Author response to Decision Letter 1]

10 Feb 2020

From the second round of reviews following primary revision, all comments and concerns were addressed from Reviewer #2, however reviewer #1 has a concern regarding the interpretation of cross-reactivity and false positives. Specifically: “In addressing the higher than expected positivity observed among pediatric subjects, the authors are correct in stating that one explanation is the presence of maternal antibodies. However, they fail to explicitly state that an alternative explanation is lack of assay specificity (i.e., pediatric positives are false positives). This should be explicitly stated in the manuscript.” We accept this as a true limitation of the current assay and an epidemiological study to identify cutoff levels would serve this assay significantly.

We have now included a statement regarding this in the manuscript in the marked-up copy on lines 476-477. We do agree that the false positives could be from other sources due to polyclonal antibody affinity and avidity. A much larger HPV-positive and -negative cohort would aid us in identifying true serostatus levels and that is a limitation of the current assay as it is presented.

---

## [Editor Report · Decision Letter 2]

12 Feb 2020

Development and validation of a multiplex immunoassay for the simultaneous quantification of type-specific IgG antibodies to E6/E7 oncoproteins of HPV16 and HPV18

PONE-D-19-34611R2

Dear Dr. Layman,

We are pleased to inform you that your manuscript has been judged scientifically suitable for publication and will be formally accepted for publication once it complies with all outstanding technical requirements.

With kind regards,

Scott M. Langevin, Ph.D., M.H.A., C.T. (ASCP)

Academic Editor

PLOS ONE
---

## [Editor Report · Acceptance letter]

13 Feb 2020

PONE-D-19-34611R2 

Development and validation of a multiplex immunoassay for the simultaneous quantification of type-specific IgG antibodies to E6/E7 oncoproteins of HPV16 and HPV18 

Dear Dr. Layman:

I am pleased to inform you that your manuscript has been deemed suitable for publication in PLOS ONE. Congratulations! Your manuscript is now with our production department. 

With kind regards,

on behalf of

Dr. Scott M. Langevin 

Academic Editor

PLOS ONE